# Analysis of Antimicrobials in Muscle and Drinking Water in Terms of Reducing the Need of Antimicrobial Use by Increasing the Health and Welfare of Pig and Broiler

**DOI:** 10.3390/antibiotics12020326

**Published:** 2023-02-04

**Authors:** Anna Gajda, Tomasz Błądek, Małgorzata Gbylik-Sikorska, Ewelina Nowacka-Kozak, Kyriacos Angastiniotis, Maro Simitopoulou, George Kefalas, Paolo Ferrari, Pierre Levallois, Christine Fourichon, Maaike Wolthuis-Fillerup, Kees De Roest

**Affiliations:** 1National Veterinary Research Institute, Partyzantow 57, 24-100 Pulawy, Poland; 2Vitatrace Nutrition Ltd., 18 Propylaion Street, Strovolos Industrial Estate, 2033 Strovolos, Cyprus; 3Nuevo Group, 32009 Schimatari, Viotia, Greece; 4Research Center for Animal Production, Viale Timavo 43/2, 42121 Reggio Emilia, Italy; 5Oniris, INRAE, BIOEPAR, 44300 Nantes, France; 6Wageningen University & Research, Animal Welfare & Adaptation, 6700 HB Wageningen, The Netherlands

**Keywords:** antimicrobials, LC-MS/MS, muscle, drinking water, reduction, pig, broiler, health, welfare

## Abstract

Antimicrobial residues may pose harmful effects on the health of consumers. At the same time, an adequate quality of drinking water for animals is one of the important element to ensure animal welfare and food without antibacterials. The presented study is aimed at estimating the residue levels of antibacterial compounds, such as penicillins, cephalosporin, macrolides, tetracyclines, quinolones, sulphonamides, aminoglycosides, diaminopirymidines, pleuromutilines and lincosamides in meat and on-farm drinking water samples using liquid chromatography-tandem mass spectrometry (LC-MS/MS), as a part of a surveillance system on pig and broiler farms within the project Healthy Livestock. A total of 870 samples of muscle from pig and broiler, as well as 229 water samples were analysed for antibiotic residues. Samples were collected from farms in EU countries in two steps, before and after implementation of a tailor-made health plan. In muscle samples, the detected concentrations of doxycycline in the post-intervention step (15.9–70.8 µg/kg) were lower than concentrations in the pre-intervention step (20.6–100 µg/kg). In water samples, doxycycline in an average concentration of 119 µg/L in the pre- and 23.1 µg/L in the post-intervention step, as well as enrofloxacin at concentrations of 170 µg/L in the pre- and 1.72 µg/L in the post-intervention step were quantified. Amoxicillin was only present before intervention. The obtained results confirm the effectiveness of the intervention actions. The concentrations of antibiotics in muscles and water were lower after implementation of a health plan on the farms.

## 1. Introduction

Over recent decades, intensive animal husbandry systems for food production have led to a significant increase in the use of veterinary medicines. To protect the health and welfare of livestock, antimicrobial agents, especially antibacterial compounds, are used worldwide, in a variety of extensive and intensive livestock production systems.

The overuse and excessive administration of antibacterials, as well as failures to comply with the warnings on antibiotic labels and withdrawal inadequacy, may cause residue occurrence in products of animal origin. Drug residues in foods derived from animals may lead to many adverse health effects for the consumer [1,2]. The residues may result in many biological adverse effects, such as allergic reactions, increased immunological responses in susceptible individuals and intestinal microbiota disturbance in consumers [3]. According to the WHO, the spread of drug-resistant bacteria, as well as bacterial resistance acquisition, is one of the major concerns for human and animal health [4,5].

The present study has focused on two selected livestock species, pigs and broilers, which are two of the top three sources of meat [6]. The consolidation of pig and broiler production requires ensuring suitable conditions of animal maintenance along with high health conditions, consistent with the guidelines for animal welfare [7]. To ensure health and welfare on farms, a suitable intervention plan needs to be elaborated. The objective of any developed biosecurity audit, such as the one described by Van Limbergen et al., 2018, as well as the one developed as part of Healthy Livestock and described by Schreuder et al., 2022, particularly in the case of intensive broiler production systems, is to identify the weak points of farm biosecurity and arrive at targeted proposals for improvements [8,9]. At the same time, antibiotic control in animals is an important element for securing higher quality animal production, while also ensuring consumer protection. To protect consumers’ health and ensure food safety with confidence in animal production, it is crucial to control all antibiotics used in pigs and broilers.

The EU has established monitoring programmes for the control of the presence of antimicrobial residues in the food chain. Regulation (EU) No 37/2010 establishes maximum residue limits (MRL) for residues of veterinary medicinal products in animal products [10]. In EU countries, the control of veterinary medicinal product residues and other substances in animal products is carried out every year, and non-complaint results are presented in annual reports. The latest report presents the results generated in 2020 as a part of official control actions. In the latest report issued from official data, only 0.14% of the samples analysed under the Directive 96/23/EC monitoring were non-compliant with antibacterial MRL in 2020, which was similar to 2019 [11,12]. According to the data included in this report, 15 countries reported a total of 42 non-compliant samples in pigs (65 non-compliant results), representing 0.12%. For antibacterials in poultry, five countries reported a total of seven non-compliant samples and results, which is 0.04%. According to European Union Reports, the percentage of non-compliant results are relatively low, but these documents present only results with concentrations above MRL values. There are some propositions and plans to report, in the future, all samples with antibiotics presence, even much lower than MRL. The EU report reveals that the most frequently used antibiotics in the pig industry in 2020 were tetracyclines and sulphonamides, while in poultry mainly tetracyclines (doxycycline) were reported. The distribution and the use of veterinary medicines in food animals is regulated by the law and responsible agencies worldwide. In compliance with the Regulation (EU) 2019/6 on veterinary medicinal products, the use of antimicrobials such as those for growth promotion and yield increase is prohibited [13].

In the face of an emerging outbreak of disease, particularly in poultry, less frequently in pigs, antimicrobial agents are added to drinking water, which is one of the most practical and economical routes of veterinary drug administration [14,15]. It provides rapid administration of medicines to all animals in the early stages of disease, low cost of solution preparation and easy distribution and drug storage, as well as facilitating quick changes of dosage [16]. However, in such a method of drugs administration, one important factor is to guarantee adequate water quality and water hygiene [17]. The physicochemical properties of drugs should be considered, including solubility in water and adsorption in the solid phase, because some substances can form complexes with the ions present in drinking water [18,19]. Contaminated water supply systems can cause the spread of medicines to the farm environment. The physicochemical properties of some antibacterials (tetracyclines, fluoroquinolones and sulphonamides) enable them to adhere to water supply system pipes and stay in the internal surface of pipes and become fixed to the biofilm [17]. This raises some issues: drugs can be systematically eluted at the end of animal treatment, causing an unintended application of antibiotics to animals. Moreover, the biofilm can in turn break away from the inner surface of the pipes and be drunk by broilers or pigs, and further spread any resistance developed among the bacteria contained within the biofilm. Additionally, after administration of various antibacterial agents in water, many interactions may take place, which can in turn disrupt the intended therapy and consequent drug elimination from body tissues. Therefore, a regular cleaning and a system of regular sanitation procedures in water supply systems with special cleaners should be implemented on each farm where food producing animals are housed. Hence one of the most important elements to ensure both animal welfare and food without antibacterial residues is the careful control of the water supply during animal production. However, in most EU countries, no official control of antibiotics in water supply systems is carried out. Published data from one study indicate the presence of antibiotics in 52% of analysed water samples [20].

The main objective of this study was to investigate the impact of the implementation of tailor-made health plans, including biosecurity measures on the results of analyses of antimicrobial residues in muscle and water. Many classes of antibiotics and antibacterial compounds, such as β-lactams (penicillins and cephalosporins), macrolides, tetracyclines, (fluoro)quinolones, sulphonamides, aminoglycosides, lincosamides, pleuromutilins and diaminopyrimidines can be administered to food-producing animals, according to EU regulation; therefore, all these groups of substances were tested in the presented study by the liquid chromatography-tandem mass spectrometry (LC-MS/MS) method. Broiler farms from the Netherlands, Cyprus and Greece and pig farms from Italy and France were involved in this research.

## 2. Results

### 2.1. Method Validation

A full validation of the methods used in this study have been previously described [20,21]. Briefly, the method for antibiotics analysis in muscle was validated according to the criteria of Commission Decision 2002/657/EC [22]. Matrix-matched calibration was used for quantification in order to reach a high accuracy. The method is linear in a wide range, as confirmed by the correlation coefficient r > 0.99, where the lowest concentration on the calibration curve refers to the limit of quantification (LOQ). The recoveries ranged from 88% to 105% and within-laboratory reproducibility was lower than 15%. The validation results of the method for determination of antibiotics in muscle and water are reported in Table 1. The matrix-matched calibration curves for water achieved good linearity (r > 0.99). The recoveries are in the range between 84% and 109%, within-laboratory reproducibilities are below 14%, and the LOQ values are in the range of 0.02–10 µg/L, depending on analyte.

### 2.2. Quantitative Analysis of Antibiotics in Muscle and Water

A total of 870 samples of muscle from pigs and broilers, as well as 229 water samples were analysed for antibiotic residues. Five hundred eighty-five muscles and 116 water samples from broiler farms were analysed. Two hundred eighty-five pig muscles and 113 water samples were tested. The main goal of this research was to demonstrate the differences in antibiotic residues before and after intervention on pig and broiler farms. No antibiotics were detected in muscle or water on pig farms, either before or after intervention. Doxycycline was detected in muscle samples from 15 broiler farms, where conventional antibiotics usage was documented and confirmed. Detected concentrations of doxycycline in the post-intervention step (15.9–70.8 µg/kg) were lower than concentrations in the pre-intervention step (20.6–100 µg/kg). Only in one muscle sample was the residue level of doxycycline equal to MRL = 100 µg/kg, before implementation of the health plan. Of all the 116 water samples analysed, antibiotics were present in 22 of them (doxycycline—15 and 13 before and after the plan, respectively; enrofloxacin—20 and 10 before and after the plan, respectively; amoxicillin—only 3 before the health plan). In water samples, doxycycline in an average concentration of 119 µg/L in the pre- and 23.1 µg/L in the post-intervention, as well as enrofloxacin in an average concentration of 170 µg/L in the pre- and 1.72 µg/L in the post-intervention step were quantified. All water samples with antibiotics were from broiler farms. The obtained results suggest the effect of the intervention actions. The concentrations of antibiotics in muscles were slightly lower after implementation of a health plan on farms. The differences in the levels of confirmed drugs are more evident in water samples. Water samples from seven farms (farm 16 to farm 22) contained 1–3 antibiotics (doxycycline, enrofloxacin or amoxicillin) before intervention, while no antibiotics in muscles from those farms were found. In these water samples, after heath plan application, only enrofloxacin with significantly lower levels was detected. At the pre-intervention step, the concentrations of amoxicillin in three water samples from three farms were relatively high (163, 2475 and 2962 µg/L), whereas antibiotics were no longer detected post-intervention. The obtained results are presented in Table 2. The statistical method used in data analysis was descriptive statistics, which summarizes data using mean concentrations and standards deviation (SD). For muscles samples (n = 10), SD was calculated, but for water samples where only two replicates were analysed, SD was not indicated. The chromatograms of LC-MS/MS analysis for water and muscle samples from broilers with detected and confirmed antibacterials are presented on Figure 1.

## 3. Discussion

Antibacterial spread began more than half a decade ago, when AMR was not considered a public health risk. The World Health Organization report from September 2021 declared AMR a major public health concern [23]. AMR is hastened by contemporary farming practices in which many animals are housed in overcrowded and unhygienic conditions that provide an ideal environments for the expansion and reproduction of antibiotic-resistant bacteria and resistance genes [24]. However, various measures are being taken to reverse the routine use of antibiotics in livestock.

The question arises: is it possible to raised animals without antibiotics? While antibiotic-free pork production is favourable, research by Dee et al., 2018, presented the problems of keeping livestock completely without antibiotics and declared them to be a serious disease challenge, such as in the case of porcine reproductive and respiratory syndrome virus [25]. While the total elimination of antibiotics can be challenging, the reduction and responsible use seems to be the priority in enhancing animal health and welfare [26]. Using antimicrobials in poultry producing meat and eggs, as well as in pigs producing meat for human consumption, should be carried out with special responsibility and attention. The prudent use of antimicrobials refers to the optimal choice of drug, dose and the time of antimicrobial treatment, along with limiting inappropriate administration and overuse. The Healthy Livestock research programme, under which the presented research was performed, is looking at a reduction in the risk of exposure of animals to pathogens; an early detection of health problems and specific diseases; increasing the resistance of animals to diseases; and if antimicrobials are necessary, a more prudent use or the application of alternatives. However, in order to reduce the use of antimicrobials in pig and broiler farms, it is important to implement biosecurity measures to prevent pathogens from entering the farms or avoid the spread of the pathogens within the farm premises. Both external and internal biosecurity measures contribute to this objective [27]

The analysis of antimicrobial residues in muscle and water on pig and broiler farms conducted as a part of the Healthy Livestock project aimed to compare antibiotic levels on pig and broiler farms before the implementation of health plans and after the use of some biosecurity measures to enhance animal health and welfare. Determination and implementation of measures were performed within other work packages. Each farm was visited to establish the weak points, in the level of either external or internal biosecurity, as identified using the BEAT risk assessment tool developed as part of the Healthy Livestock project [9]. Based on this analysis, tailor made health plans were designed. The biosecurity measures implemented on broiler and pig farms are presented in the Appendix A. The farm selection was an important factor in the implementation of the intervention plans and the influence on antibiotic residues detection. The broiler farms selected, particularly in Cyprus and Greece, but not so in the Netherlands, were known to be reliant on the use of antibiotic veterinary interventions in previous production cycles. Although they were not typical of the specific countries, they were specifically selected for this study so as to best test the hypothesis as to whether improvements in biosecurity would indeed improve the health status of subsequent production flocks and hence reduce the need of veterinary interventions and consequently the use of antibiotics. The BEAT system was tested on farms in Cyprus, Greece and the Netherlands to asses if the implementation of the health plan resulted in a reduction in antimicrobial use. In improving the health status of animals and reducing the use of antibiotics, thus decreasing the incidence of antibiotic residues, short-term and low-cost interventions that were mostly aimed at improving disinfection and the training of people entering the farm have proven to be most effective [9]. A detailed description of the results of implementation of health plans in broiler farms is presented in Schreuder et al., 2022 [9]. In pig farms, no residues were confirmed.

The results obtained in this research show that the percentage of muscle samples with antibacterials was quite low and reached 2.6%, considering only chicken muscles in general. In water samples, doxycycline was quantified at the pre- and post-intervention step as 13.7% and 11.2%, respectively. Even though the obtained percentage difference is slight, the concentrations of antibiotic were much lower after biosecurity actions. For enrofloxacin, both percentages of positive samples (17.2% pre- and 8.6% post-intervention) as well as concentrations were significantly lower, after biosecurity was undertaken. Samples of water with amoxicillin were found at 1.7% before interventions, while all samples were negative after the measures were introduced. On pig farms, no antibiotics were detected, both before and after implementation of the health plan.

In the presented study, doxycycline was found in both muscle and water, both pre- and post-intervention, which indicates the high stability of this antibiotic [28]. Despite the presence of enrofloxacin and amoxicillin residues in a few water samples, no residues in muscles were confirmed. Similar to the results presented in this paper, the most frequently found compound in muscles from the group of tetracyclines in the EU is doxycycline [11,12]. According to an EU Report in 2019, doxycycline was found in 24 muscle samples of pigs and 6 samples in poultry. Amoxicillin was confirmed only in pigs (six non-compliant results), while enrofloxacin was detected in one muscle sample in poultry. In a 2020 EU report, most non-compliant results in pigs concerned tetracyclines (28 samples), including 12 results with doxycycline. From (fluoro)quinolones, enrofloxacin was presented in five samples, while for penicillin, amoxicillin was confirmed in two samples together with penicillin G in three samples. In poultry, only seven non-compliant results were found, and four of these were related to doxycycline. According to the ranking of antibiotic families based on their occurrence (%) created by the World Organization of Animal Health (OIE), the most widely used group of antibacterials are tetracyclines (87.1%) and penicillins (87.1%) [29].

Roblez-Jimenez et al., 2022, reported the concentration of antibiotic residues found in the environment, livestock, animal tissues, animal products (milk and eggs), wastewater and soil, based on a very comprehensive literature review [30]. According to that study, the levels of antibiotics based on continent showed a notable differences among antimicrobials groups [30]. The antimicrobial with the highest concentration in Asia was cephalosporin, followed by fluroquinolone. The highest residual concentrations in Africa and North America involved tetracyclines, while in South America fluoroquinolones and macrolides were the most frequent. In Europe, the highest concentrations were shown by β-lactam; however, in Europe the main antibiotics sold were tetracyclines (32.8%), penicillins (25.0%) and sulphonamides (11.8%) [31]. Considering the residues of antibiotics in animal products, based on the data from various parts of the world, the largest concentration of residues was found in chicken, with the main occurrences being fluoroquinolones and tetracyclines [30].

The residue of antibiotics on farms can be present due to animal excretion, pig and poultry faeces or manure. In the literature data, the most commonly detected antibiotics in manure, faeces and slurry are tetracyclines [32]. In the research of Patyra et al., 2020, out of 70 pig and poultry faeces and manure samples, 15 were positive for doxycycline [33]. According to Rasschaer et al., 2020, the most frequently detected antibiotics were doxycycline, sulfadiazine and lincomycin, but doxycycline was found in the highest concentration, with a mean of 1476 μg/kg manure [34]. Residues of some antibiotics in poultry can also be present in feathers. Gajda et al., 2019, demonstrated high concentrations of doxycycline in broiler feathers for a long time after (22 days) post-treatment [35].

## 4. Materials and Methods

### 4.1. Sample Collection on Broiler and Pigs Farms

Muscle and water samples were collected from broiler and pig farms. Thirteen broiler houses from the Netherlands, seven from Cyprus and ten from Greece were recruited to participate in this study. Twenty pig houses in France and fifteen pig farms in Italy were also involved. All pig and broiler farms were identified and documented. The objective was to identify the biosecurity and health standards on the site and what were considered the key areas needing improvement, with the aim of decreasing the need for antimicrobial use while maintaining biological and economic performance. Such health plans included changes which could mitigate risks, could be easily implemented and could reduce the use of antimicrobials. Biological and economic data, as well as antimicrobial use, were recorded for each farm in pre-intervention cycles and post-intervention cycles. Samples for detecting any residues in muscle and water were collected pre- and post-intervention. The number of samples collected from reach country is listed in Table 3. In France, it was not possible to collect muscle samples, as no access was provided by abattoirs due to the COVID-19 pandemic.

#### 4.1.1. Farm Selection

Broiler and pig farms were selected based on the following criteria:Farms had to be users of antimicrobials if any progress on this aspect was to be demonstrated before any health plan implementation. Because it was difficult to recruit farms in the Netherlands, farms with no antimicrobial usage were also included.Farm veterinarians and farmers had to be willing to be involved.Participant farms ideally had to cover a range of pig and broiler houses and practices in place, including the age of the buildings, house equipment such as feeding systems and type of bedding, as well as the labour employed.In addition, broiler and pig density and other commercial livestock pressures on the location of each farm had to be considered so as to have a representative range of farm locations.

Each broiler house was based on a different farm, except in Greece where some broiler houses were within the same farm area. These broiler houses had different management and antimicrobial histories and thus were handled as independent houses. For the Netherlands, only one broiler house per farm was sampled, but on some farms, the biological data of multiple houses was collected per farm. Depending on the farm (broiler or pig), the criteria for the monitoring of pre- and post-intervention flock cycle monitoring were different. Four production cycles (cycles 1–4) were followed up, of which two cycles were considered as pre-intervention and two cycles as post-intervention. Water samples were taken at the end of rounds 1 (first) and 4 (last). Meat samples were collected at the slaughterhouse after the first and last round. Between rounds 2 and 3, the intervention plan was made and, depending on the sort of intervention, it was done at round 4.

#### 4.1.2. Pre-Intervention Flock Cycle Monitoring

It was agreed that two broiler flock cycles were first to be monitored, where the active collection of data, as well as sampling for antimicrobial residues in targeted material and selected biomarker scores, were to be recorded, before any health plan implementation (intervention) took place. A protocol to monitor risk mitigation in pig farms during a 12-month study period was developed; at least three visits were performed in each farm. Antimicrobial residues were monitored by collecting the following representative samples:Water at the end of the water line towards the end of the cycle when the first of the broilers were selected at thin-out. Approximately 200 mL of water were collected per occasion. These were stored a −20 °C until dispatched to the National Veterinary Research Institute in Poland for residue analysis.Muscle at the processing plant at or near the first thinning. For this, five birds were sampled and combined into one. These were also stored at −20 °C and then dispatched to the National Veterinary Research Institute in Poland for residues analysis. LC-MS/MS analyses were performed up to 1 week after receiving the samples.

Complete details of any antimicrobial usage, including age of birds, details of the vet prescription, the pharmaceutical product used and dosage, as well as the dates of administration were also recorded, but that was part of another work package.

#### 4.1.3. Post-Intervention Flock Cycle Monitoring

In broilers, following the sanitary vacuum before the second pre-intervention cycle, the first post-intervention cycle of monitoring started. This was followed in all cases by a second post-intervention period. In pigs, the interval between the pre-intervention cycle and the post-intervention cycle lasted from 8 to 12 months. This interval enabled the implementation of the tailor-made health plan between the different batches.

The same monitoring and sampling process as carried out before for the pre-intervention cycles was also repeated for all post-intervention broiler flocks and pig batches. This facilitated the opportunity to compare the outcomes of the health plans which were implemented, within the time constraints imposed, with a before and after effect, with each farm site being its own control.

### 4.2. LC-MS/MS Analysis

#### 4.2.1. Chemicals and Reagents

Reagents. All organic solvents were HPLCgrade and all chemicals were analytical grade. Acetonitrile, was from J.T. Baker (Deventer, the Netherlands). Trichloroacetic acid (TCA) and sodium acetate was from Sigma-Aldrich (St. Louis, MO, USA). Heptafluorobutyricacid (HFBA) was from Fluka (St. Louis, MO, USA). PVDF filters were from Restek (College, PA, USA). Strata X columns were form Phenomenex (Torrance, CA, USA). Water was deionised (>18 MΩ cm^−^^1^) in-house by the Millipore system.

Analytical standard and standard solutions. Amoxicillin (AMOX), ampicillin (AMPI), penicillin G (PEN G), penicillin V (PEN V), oxacillin (OXA), cloxacillin (CLOX), nafcillin (NAF), dicloxacillin (DICLOX), cephapirin (CFPI), ceftiofur (CFT), cefoperazone (CFPE), cephalexin (CFLE), cefquinome (CFQ), cefazolin (CFZ), cefalonium (CFLO), sulfaguanidine (SGU), sulfadiazine (SDZ), sulfathiazole (STZ), sulfamerazine (SME), sulfamethazine (SMT), sulfamethoxazole (SMA), sulfamethoxypyridazine (SMP), sulfamonomethoxine (SMM), sulfadoxine(SDX), sulfaquinoxaline (SQX), sulfadimethoxine(SDMX), tylosin (TYL), erythromycin (ERY), spiramycin (SPI), tilmicosin (TIL), josamycin (JOS), danofloxacin (DAN), difloxacin (DIF), enrofloxacin(ENR), ciprofloxacin (CIP), flumequine (FLU), sarafloxacin (SAR), marbofloxacin (MAR), norfloxacin(NOR), oxolinic acid (OXO), nalidixic acid (NAL), chlortetracycline (CTC), tetracycline (TC), doxycycline(DC), oxytetracycline (OTC), streptomycin (STRP), dihydrostrepromycin (DISTRP), gentamycin (GEN),paromomycin (PAR), spectinomycin (SPEC), kanamycin (KAN), neomycin (NEO), lincomycin (LIN) and sulfaphenazole (IS) were from Sigma-Aldrich.

#### 4.2.2. LC-MS/MS Analysis of Muscle

Chicken or pig meat samples were minced, homogenized and stored at 0 °C until analysis. These samples were analysed for the presence of 57 antimicrobial drugs by the LC-MS/MS method, using two extraction methods previously described by Błądek et al. [21].

Briefly, the first extraction method by acetonitrile is suitable for detecting and quantifying 45 antibiotics belonging to the following classes: β-lactams, sulphonamides, macrolides, fluoroquinolones, pleuromutilins and diaminopyrimidines. The protocol of the first method was as follows. To a muscle subsample (2 g), 8 mL of acetonitrile was added, mixed thoroughly and centrifuged. Then, 6 mL of supernatant was taken and evaporated to dryness at 45 °C. The dry residue was dissolved in 0.6 mL of 0.025% heptafluorobutyric acid (HFBA) and filtered through a 0.22 μm PVDF filter into a LC vial.

The second extraction method by aqueous solution of 5% trichloroacetic acid (TCA) allows the isolation of 12 antibiotics (aminoglycosides, tetracyclines, lincosamides). Extraction with this method involved adding 6 mL of 5% TCA to 2 g of muscle subsample. The sample was vortex mixed and centrifuged. Finally, the TCA extract (1 mL) was taken and filtered by a 0.22 um PVDF filter to vial for LC-MS/MS analysis.

LC-MS/MS analysis was performed by the Agilent 1200 HPLC system (Agilent Technologies, Santa Clara, CA, USA) connected to an API 4000 triple quadrupole mass spectrometer (AB Sciex Framingham, MA, USA). Separation of target compounds was performed on a Luna C18 (2) 100 A column (150 × 2.0 mm, 3 μm) using acetonitrile (A) and 0.025% HFBA (B) as mobile phases in gradient mode [21]. For quantification, two product ions were monitored to ensure specific and accurate quantification. The information on ion transitions and optimal conditions for the fragmentation of monitored antibiotics are provided in Table 4.

#### 4.2.3. LC-MS/MS Analysis of Water

Water from breeding animal watering supply was analysed by LC-MS/MS, as previously described by Gbylik-Sikorska et al., 2015 [20]. Forty-five veterinary compounds belonging to nine different antibiotic groups, including aminoglycosides, β-lactams, diaminopyrimidines, fluoroquinolones, lincosamides, macrolides, pleuromutilins, sulphonamides and tetracyclines, were determined. The tested antibiotics are marked with an asterisk in Table 1.

Isolation of antimicrobial substances from the water samples was based on extraction with sodium acetate and the addition of ionic pairs, followed by solid phase extraction (SPE) [20]. Concisely, to 250 mL of water, 6 mL of 0.5 M sodium acetate, pH = 5.6, and 30 µL of HFBA were added, and the sample was shaken briefly for 5 min. Next, the sample was transferred to a conditioned Strata-X SPE column. The analytes were eluted from the SPE with 3 mL of a mixture of acetonitrile: 0.05 M HFBA (9:1, *v/v*), and the eluate was evaporated to dryness. The dry residue was dissolved in 500 ul of 0.025% HFBA and filtered through 0.22 um PVDF syringe filters into LC vials.

LC-MS/MS analysis of water was performed on the same instrument as the meat sample. However, chromatographic separation of analytes was performed on a Luna C18 (2) 100 A column (50 × 3.0 mm, 3 µm) using the same mobile phases but in a different gradient mode.

## 5. Conclusions

In modern livestock production systems, efforts to maintain high health standards may imply some use of antimicrobial drugs in farm animals. The mean objective of the presented study was to reduce the use of antimicrobials administrated on pig and broiler farms by implementing tailor-made health plans, including biosecurity measures, and to investigate the possible change in residues in water and meat samples. The results obtained in this study indicate a reduction in antibiotic residues in water samples on broiler farms when biosecurity measures were improved. No residues were found in the samples from pig farms. The research on antimicrobials reduction by the implementation of selected intervention actions on animal farms needs to be continued and further improved.

## Figures and Tables

**Figure 1 antibiotics-12-00326-f001:**
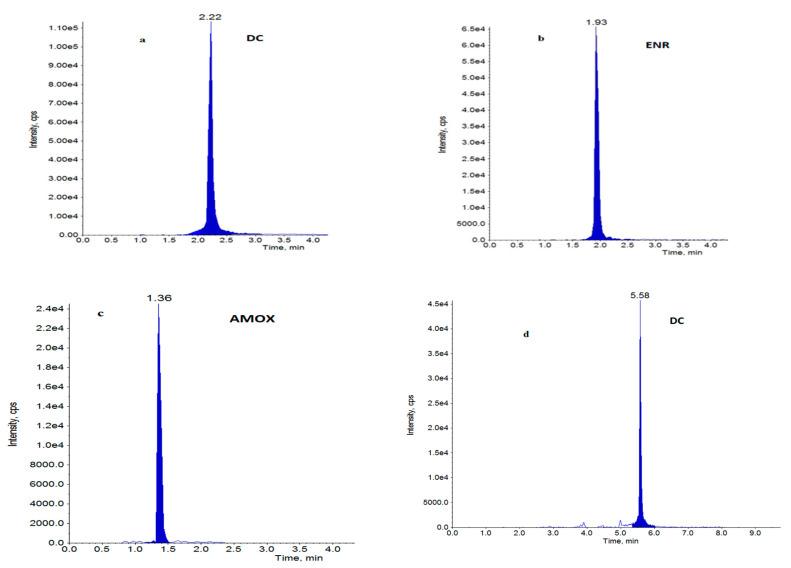
Chromatograms of water samples (**a**–**c**) with: (**a**) doxycycline, (**b**) enrofloxacin, (**c**) amoxicillin at a concentration of 5 µg/L and muscle sample, (**d**) with doxycycline at a concentration of 5 µg/kg.

**Table 1 antibiotics-12-00326-t001:** Recoveries, reproducibilities and LOQs achieved during validation study on spiked muscle and water.

Analyte	Muscle			Water		
	Recovery (%)	Reproducibility (%)	LOQ (μg/kg)	Recovery (%)	Reproducibility (%)	LOQ (μg/L)
Amoxycillin *	98	14.4	2	98	4.0	10
Ampicillin *	100	11.5	2	106	12.2	0.05
Penicillin G *	102	14.3	2	97	12.7	10
Penicillin V	99	13.1	2	-	-	-
Oxacillin *	98	7.2	2	90	9.3	0.05
Cloxacillin	100	7.0	2	-	-	-
Nafcillin *	101	6.7	2	100	9.2	0.05
Dicloxacillin *	100	7.0	2	105	8.7	0.05
Cephapirin *	92	9.3	25	92	13.6	0.05
Cefoperazone *	98	7.3	25	91	9.9	0.02
Cephalexin *	100	8.6	50	93	13.9	0.05
Cefquinome *	91	13.9	10	103	9.7	0.02
Cefazolin *	100	10.0	25	99	7.9	0.02
Cefalonium *	99	11.5	10	96	11.2	0.02
Ceftiofur *	98	4.6	50	103	8.8	0.05
Sulfaguanidine	88	9.3	5	-	-	-
Sulfadiazine	92	5.9	5	-	-	-
Sulfathiazole *	93	7.3	5	105	7.9	0.02
Sulfamerazine *	97	6.3	5	104	11.4	0.02
Sulfamethazine *	97	6.6	5	107	10.1	0.02
Sulfamethoxazole *	98	5.8	5	102	7.0	0.02
Sulfamethoxypyridazine	97	6.6	5	-	-	-
Sulfamonomethoxine *	99	7.3	5	91	9.1	0.02
Sulfadoxine	99	7.6	5	-	-	-
Sulfadimethoxine *	100	7.7	5	99	8.1	0.02
Sulfaquinoxaline	99	10.0	5	-	-	-
Trimethoprim *	99	9.6	5	95	12.5	0.05
Tylosin *	95	10.4	5	91	10.5	0.02
Erythromycin *	97	11.8	5	96	9.4	5
Spiramycin *	99	12.1	5	96	8.0	0.05
Tilmicosin *	99	13.1	5	103	12.2	0.05
Josamycin *	100	10.6	5	104	11.3	0.05
Tulathromycin	100	5.5	10	-	-	-
Danofloxacin *	95	11.6	5	96	8.3	0.02
Difloxacin *	98	12.0	5	95	6.1	0.02
Enrofloxacin *	98	10.3	5	89	7.1	0.02
Ciprofloxacin *	93	13.6	5	88	10.5	0.02
Flumequine *	105	11.9	5	102	10.2	0.02
Sarafloxacin *	100	13.9	5	84	9.1	0.02
Marbofloxacin *	93	9.3	5	86	10.9	0.02
Norfloxacin *	92	11.2	5	85	11.6	0.02
Oxolinic acid *	102	12.1	5	105	9.1	0.02
Nalidixic acid *	100	11.5	5	109	11.1	0.02
Tiamulin *	98	7.6	1	97	7.3	0.02
Valnemulin	100	11.8	5	-	-	-
Chlortetracycline *	97	13.1	5	99	10.5	0.05
Tetracycline *	100	13.6	5	96	8.0	0.05
Doxycycline *	100	13.2	5	96	13.3	0.05
Oxytetracycline *	96	14.0	5	99	12.9	0.02
Streptomycin *	96	9.7	25	96	7.1	1
Dihydrostreptomycin *	95	10.6	25	91	7.6	2
Gentamycin	99	13.2	25	-	-	-
Paromomycin	97	9.7	250	-	-	-
Spectinomycin *	96	11.4	100	94	7.6	1
Kanamycin	95	11.2	50	-	-	-
Neomycin *	102	11.3	250	97	6.8	10
Lincomycin *	95	10.2	5	99	8.5	0.02

* analytes tested in water.

**Table 2 antibiotics-12-00326-t002:** Average concentrations of antibiotics detected in muscle and water samples per broiler farm.

Farm	Muscle		Water					
	Doxycycline (µg/kg) Mean ± SD		Doxycycline (µg/L) Mean		Enrofloxacin (µg/L) Mean		Amoxicillin (µg/L) Mean	
	Pre-Intervention	Post-Intervention	Pre-Intervention	Post-Intervention	Pre-Intervention	Post-Intervention	Pre-Intervention	Post-Intervention
1.	80.0 ± 33.6	70.8 ± 36.3	13.1	6.3	6.3	ND	ND	ND
2.	30.7 ± 5.7	28.2 ± 6.9	315	1.8	20.4	ND	ND	ND
3.	42.2 ± 14.0	30.1 ± 13.2	42	ND	2.5	ND	ND	ND
4.	39.9 ± 11.4	35.6 ± 16.2	18.2	5.8	0.4	0.7	ND	ND
5.	23.6 ± 2.7	49 ± 24.3	14.1	ND	0.2	ND	ND	ND
6.	20.6 ± 1.6	20 ± 4.6	21.4	2.0	4.3	0.1	ND	ND
7.	24.4 ± 7.8	20.8 ± 13.1	40.5	2.1	7.7	ND	ND	ND
8.	20.8 ± 7.0	40 ± 18.5	405	216	3.8	9.3	ND	ND
9.	64.7 ± 37.1	64.8 ± 26.8	66	0.8	3.4	0.8	ND	ND
10.	67.6 ± 33.4	39 ± 22.7	15.6	0.5	0.3	0.05	ND	ND
11.	100 ± 31.4	34.4 ± 11.5	28.3	3.5	ND	ND	ND	ND
12.	49.2 ± 11.2	23.6 ± 7.1	64.9	1.3	1.0	ND	ND	ND
13.	83.4 ± 17.7	48.6 ± 32.6	42.6	0.9	0.3	ND	ND	ND
14.	36.5 ± 5.3	25.1 ± 8.5	774	12.6	20	0.6	ND	ND
15.	29.3 ± 11.2	15.9 ± 10.4	7.3	0.3	ND	0.2	ND	ND
16.	ND	ND	ND	ND	1.2	1.2	ND	ND
17.	ND	ND	12.5	ND	1880	ND	ND	ND
18.	ND	ND	ND	ND	0.7	ND	2475	ND
19.	ND	ND	ND	ND	49.2	0.08	ND	ND
20.	ND	ND	ND	ND	35.5	ND	163	ND
21.	ND	ND	ND	ND	13.3	ND	2926	ND
22.	ND	ND	ND	ND	5.1	0.2	ND	ND

ND—not detected.

**Table 3 antibiotics-12-00326-t003:** Number of muscle and water samples collected for antimicrobial residue analysis.

**Muscle**					
	Broiler farms			Pig farms	
Country	NL	GR	CYP	FR	IT
Samples (total)	125	320	140	0	285
**Water**					
	Broiler farms			Pig farms	
Country	NL	GR	CYP	FR	IT
Samples (total)	24	64	28	72	41

**Table 4 antibiotics-12-00326-t004:** List of analytes and mass spectrometry parameters for detection of antibacterial compounds.

Class	Analyte	Precursor (m/z)	Products 1/2 (m/z)	DP (V)	CE 1/2 (V)
β-lactams	Amoxycillin *	366	349/208	45	14/18
	Ampicillin *	350	106/160	58	27/19
	Penicillin G *	335	160/176	60	17/19
	Penicillin V	351	160/114	54	17/48
	Oxacillin *	402	160/243	50	18/18
	Cloxacillin	436	160/277	50	20/20
	Nafcillin *	415	199/171	50	20/50
	Dicloxacillin *	470	160/311	50	20/20
	Cephapirin *	424	154/124	50	35/70
	Cefoperazone *	646	530/143	60	17/50
	Cephalexin *	348	158/106	50	10/23
	Cefquinome *	529	134/125	50	25/75
	Cefazolin *	455	323/156	50	15/23
	Cefalonium *	459	337/152	46	16/28
	Ceftiofur *	524	241/125	50	25/70
Sulphonamides	Sulfaguanidine	215	156/108	20	20/30
	Sulfadiazine	251	156/108	53	22/30
	Sulfathiazole *	256	156/108	53	20/34
	Sulfamerazine *	265	156/108	45	25/37
	Sulfamethazine *	279	156/108	50	25/36
	Sulfamethoxazole *	254	156/108	50	23/35
	Sulfamethoxypyridazine	281	156/108	60	25/35
	Sulfamonomethoxine *	281	156/108	50	23/37
	Sulfadoxine	311	156/108	60	25/40
	Sulfadimethoxine *	311	156/108	50	23/37
	Sulfaquinoxaline	301	156/108	50	23/40
Diaminopyrimidines	Trimethoprim *	292	231/262	52	33/36
Macrolides	Tylosin *	916	174/772	110	52/42
	Erythromycin *	734	158/576	75	42/27
	Spiramycin *	843	174/540	120	52/44
	Tilmicosin *	869	174/696	135	61/56
	Josamycin *	828	174/229	80	46/44
	Tulathromycin	806	577/158	95	37/59
(Fluoro)quinolones	Danofloxacin *	358	340/255	60	33/50
	Difloxacin *	400	382/356	50	30/28
	Enrofloxacin *	360	342/286	72	30/50
	Ciprofloxacin *	332	314/231	61	30/47
	Flumequine *	262	244/202	44	25/45
	Sarafloxacin *	386	368/348	50	31/46
	Marbofloxacin *	363	345/320	70	30/22
	Norfloxacin *	320	302/231	60	33/50
	Oxolinic acid *	262	244/216	53	25/40
	Nalidixic acid *	233	215/187	42	30/35
Pleuromutilines	Tiamulin *	494	192/118	128	30/56
	Valnemulin	565	263/164	45	20/40
Tetracyclines	Chlortetracycline *	479	444/462	56	31/25
	Tetracycline *	445	410/427	36	27/19
	Doxycycline *	445	428/154	55	25/42
	Oxytetracycline *	461	426/443	41	27/19
Aminoglycosides	Streptomycin *	582	263/246	166	45/52
	Dihydrostreptomycin *	584	263/246	150	42/53
	Gentamycin	478	322/157	44	22/31
	Paromomycin	616	163/293	112	49/33
	Spectinomycin *	351	333/207	67	27/32
	Kanamycin	485	163/205	70	35/36
	Neomycin *	615	161/163	109	46/33
Lincosamides	Lincomycin *	407	126/359	74	36/28

* analytes tested in water.

## Data Availability

All data are contained within the article.

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
