# Peer review of "Analysis of Antimicrobials in Muscle and Drinking Water in Terms of Reducing the Need of Antimicrobial Use by Increasing the Health and Welfare of Pig and Broiler"

_antibiotics, 2023, doi:10.3390/antibiotics12020326_

Round 1

Reviewer 1 Report

Dear Authors,

The manuscript “Analysis of antimicrobials in muscle and drinking water in terms of reducing the need of antimicrobials use by increasing the health and welfare in pigs and poultry” quantified a wide range of animal antimicrobial drugs residues by HPLC-MS/MS method in pig and poultry muscle and water samples from different EU countries before and after the intervention of animal health plan. A total of 870 samples of muscle 24 from pig and poultry, as well as 229 water samples, were analysed and found and only doxycycline was found in broiler muscle while no drug residue was found in pig muscle. For both muscle and water, the drug residues were significantly reduced after the implementation of the animal intervention plan. This was a well-planned big research job during the pandemic situation. The analytical procedure was well described with good method validation. However, there needs some rewriting in the introduction and description sections. A minor revision is recommended.

General comments:

1. In the Introduction section, add previously published literature with results about antimicrobial residue in animal and poultry muscle and water with reference.

2. In the introduction, briefly add the EU intervention plan for animal health with reference and their relation with drug residues.

3. Remove unnecessary literature from the Introduction section.

4. In discussion, correlate your findings with recently published drug residue data in the EU and other regions with explanations.

5. In discussion, correlate your residue findings with the implementation of the Intervention plan for animal health.

6. Remove unnecessary literature from the discussion chapter.

7. Add a reference for every literature cited in your introduction and discussion chapter. 

Author Response

Dear Reviewer

Thank you for your valuable comments, meticulous review, as well as professional suggestions. We have made revisions of the manuscript according to your suggestions. We hope that the revised manuscript meet your requirements.

The manuscript “Analysis of antimicrobials in muscle and drinking water in terms of reducing the need of antimicrobials use by increasing the health and welfare in pigs and poultry” quantified a wide range of animal antimicrobial drugs residues by HPLC-MS/MS method in pig and poultry muscle and water samples from different EU countries before and after the intervention of animal health plan. A total of 870 samples of muscle 24 from pig and poultry, as well as 229 water samples, were analysed and found and only doxycycline was found in broiler muscle while no drug residue was found in pig muscle. For both muscle and water, the drug residues were significantly reduced after the implementation of the animal intervention plan. This was a well-planned big research job during the pandemic situation. The analytical procedure was well described with good method validation. However, there needs some rewriting in the introduction and description sections. A minor revision is recommended.

General comments:

  1. In the Introduction section, add previously published literature with results about antimicrobial residue in animal and poultry muscle and water with reference.

Authors: The most reliable and comprehensive source of data on antimicrobial residue in pig and poultry muscle are European Union reports on monitoring of veterinary medicinal product residues and other substances in live animals and animal products, carried out every year. The authors cited the results of a control study in EU. “According to latest reports, for antibacterials, 15 countries reported a total of 42 non-compliant samples in pigs (65 non-compliant results), representing 0.12 %. For antibacterials in poultry, five countries reported a total of seven non-compliant samples and results, which is 0.04%. EU report reveal that the most frequently used antibiotics pig industry in 2020 are tetracyclines and sulphonamides, while in poultry mainly tetracyclines (doxycycline) were reported.”

Additionally, in discussion section some literature and information about most often detected group of antibacterials was updated.

There is no data in the literature about antimicrobial residues in water from water supply system in food producing animals farms, with the exception of one publication which was quoted (Gbylik et al. , 2015).

  1. In the introduction, briefly add the EU intervention plan for animal health with reference and their relation with drug residues.

Authors: It was completed in the Introduction part.  

  1. Remove unnecessary literature from the Introduction section.

Authors: Some literature was removed. Update of references was done.

  1. In discussion, correlate your findings with recently published drug residue data in the EU and other regions with explanations.

Authors: It was completed in discussion section.

  1. In discussion, correlate your residue findings with the implementation of the Intervention plan for animal health.

Authors: The biosecurity measures implemented on broiler and pig farms are presented in Supplementary Material (Table S1 and Table S2). On each farm the different interventions were implemented. The detailed description of Intervention plan and implementation was described in Schreader et al. paper (Poultry Science, 2023, 102). In discussion briefly comments was completed.     

  1. Remove unnecessary literature from the discussion chapter.

Authors: The discussion was generally revised and improved, some unnecessary references and fragments was removed.

  1. Add a reference for every literature cited in your introduction and discussion chapter. 

Authors: It was completed.

Reviewer 2 Report

1. A brief summary

     The aim of the paper entitled Analysis of antimicrobials in muscle and drinking water in terms of reducing the need of antimicrobials use by increasing the health and welfare in pigs and poultry is appropriate for consideration of publication in the Antibiotic. This highlighted outcome could be used to positively improve the antimicrobial stewardship and the minimizing the use of antimicrobials for intensive livestock husbandry systems.

2. General concept comments

2.1 The manuscript is moderately structured and written. It has a crucial clinical message and ought to pique the readers' interest. Unfortunately, the citing recent references also published within the last 5 years is 38.24% (13/34 publications). Please consider updating the references.

2.2 The unspecific title of this paper is insufficiently reflect its contents. I am pleased to suggest that the term "broiler" should be used instead of "poultry."

2.3 To be consistent with the experimental sample, the word "broiler" should be also used in place of the word "poultry" in the abstract. 

2.4 Please append the evidence references for line 53-54 The present study has focused on two selected livestock species, pigs and broilers, which globally provide two of the top three sources of meat.”

2.5 For method of farm selection, you need specify time (and/or distance) data for sampling. It is important for comprehending the occurrence and distribution of antibiotic residues.

2.6 Please provide instructions on the statistical analysis and data management that were carried out for the methodology section.

2.7 I advise that you must improve the discussion section's description. The analysis and interpretation of each experiment's findings should be done in this part. In addition, the result on "Method validation" have not received any discussion.

2.8 Since, the final paragraph of the discussion section just repeats the trial results. Your discussion section needs more detail.

2.9 The conclusion section of this manuscript is quite vague and lengthy. Some conclusions should state the primary point of the obtained major content and make recommendations for clinical practice. Additionally, this research's conclusion section requires a suggestion for further research.

Author Response

Dear Reviewer

Thank you for your valuable comments, meticulous review, as well as professional suggestions. We have made revisions of the manuscript according to your suggestions. We hope that the revised manuscript meet your requirements.

Comments and Suggestions for Authors

  1. 1.A brief summary

     The aim of the paper entitled “Analysis of antimicrobials in muscle and drinking water in terms of reducing the need of antimicrobials use by increasing the health and welfare in pigs and poultry” is appropriate for consideration of publication in the Antibiotic. This highlighted outcome could be used to positively improve the antimicrobial stewardship and the minimizing the use of antimicrobials for intensive livestock husbandry systems.

  1. 2.General concept comments

2.1 The manuscript is moderately structured and written. It has a crucial clinical message and ought to pique the readers' interest. Unfortunately, the citing recent references also published within the last 5 years is 38.24% (13/34 publications). Please consider updating the references.

Authors: References were updated. Some literature was removed. All new positions added to the references (10), were published within the last 5 years.

2.2 The unspecific title of this paper is insufficiently reflect its contents. I am pleased to suggest that the term "broiler" should be used instead of "poultry."

Authors: Thank you for this suggestion. It was changed.

2.3 To be consistent with the experimental sample, the word "broiler" should be also used in place of the word "poultry" in the abstract. 

Authors: Of course, it was corrected in all manuscript.

2.4 Please append the evidence references for line 53-54 “The present study has focused on two selected livestock species, pigs and broilers, which globally provide two of the top three sources of meat.”

Authors: The reference was added.

2.5 For method of farm selection, you need specify time (and/or distance) data for sampling.  It is important for comprehending the occurrence and distribution of antibiotic residues.

Authors:  Four production cycles (cycles 1–4) were followed up, of which 2 cycles were considered as pre intervention and 2 cycles as post intervention. Water samples were taken at the end of round 1 (first) and 4 (last). Meat samples were collected at the slaughterhouse after the first and last round. Between round 2 end 3 the intervention plan was made and depending on the sort of intervention it was done at round 4

2.6 Please provide instructions on the statistical analysis and data management that were carried out for the methodology section.

Authors: The statistical method used in data analysis was descriptive statistics, which summarizes data using mean concentrations and standards deviation (SD). Mean concentrations with SD in Table 2 were completed.

2.7 I advise that you must improve the discussion section's description. The analysis and interpretation of each experiment's findings should be done in this part. In addition, the result on "Method validation" have not received any discussion.

Authors: The discussion was generally revised and improved. The methods were not the main object of the presented paper, it was only analytical tool in this research. Presented methods were previously published, but the validation results were briefly discussed  in point 2.1.

2.8 Since, the final paragraph of the discussion section just repeats the trial results. Your discussion section needs more detail.

Authors: The results in this section are discussed with regard to total number of analyzed muscle and water samples in this study. Samples with antibiotics are presented in %. Discussion section was generally revised.

2.9 The conclusion section of this manuscript is quite vague and lengthy. Some conclusions should state the primary point of the obtained major content and make recommendations for clinical practice. Additionally, this research's conclusion section requires a suggestion for further research.

Authors: Conclusion section was changed and shortened.

Reviewer 3 Report

This study aimed to estimate the residue levels of antibacterial compounds, such as penicillins, cephalosporin, macrolides, tetracyclines, quinolones, sulfonamides, aminoglycosides, diaminopirymidines, pleuromutilines and lincosamides in meat and drinking water samples using LC-MS/MS.

The main goal of this research was to demonstrate the differences in antibiotic usage before and after intervention on pig and poultry farms. The study sample range is good, with 870 samples of muscle, 24 from pig and poultry, and 229 water samples

The method accuracy is high. The method is linear in a wide range as confirmed by the correlation coefficient r > 0.99; the lowest concentration on the calibration curve refers to the limit of quantification (LOQ).

Firstly, the paper is well organized; by the way, my suggestions are below;

The authors represented the result as a report Project results I recommended to the authors represented as a paper result. Discussion parts need to improve

Author Response

Dear Reviewer

Thank you for your valuable comment and professional suggestions. We have made revisions of the manuscript according to your suggestion. We hope that the revised manuscript meet your requirements.

This study aimed to estimate the residue levels of antibacterial compounds, such as penicillins, cephalosporin, macrolides, tetracyclines, quinolones, sulfonamides, aminoglycosides, diaminopirymidines, pleuromutilines and lincosamides in meat and drinking water samples using LC-MS/MS.

The main goal of this research was to demonstrate the differences in antibiotic usage before and after intervention on pig and poultry farms. The study sample range is good, with 870 samples of muscle, 24 from pig and poultry, and 229 water samples

The method accuracy is high. The method is linear in a wide range as confirmed by the correlation coefficient r > 0.99; the lowest concentration on the calibration curve refers to the limit of quantification (LOQ).

Firstly, the paper is well organized; by the way, my suggestions are below;

The authors represented the result as a report Project results I recommended to the authors represented as a paper result. Discussion parts need to improve

Authors: Discussion part was generally revised and improved.